# Severe outcomes of COVID-19 among adults with increased risk conditions: A population-based observational study

Scott Dryden-Peterson[1,2,3*], Andy Kim[1], Ellen C. Caniglia[4], Mary-Ruth Joyce[1],
David Rubins[1,2,3], Arthur Y. Kim[3,5], John Fangman[6], Lindsey R. Baden[1,3],
Ann E. Woolley[1,3]

1 Brigham and Women's Hospital, Boston, Massachusetts, United States of America, 2 Mass General Brigham, Somerville, Massachusetts, United States of America, 3 Harvard Medical School, Boston, Massachusetts, United States of America, 4 Perelman School of Medicine, University of Pennsylvania, Philadelphia, Pennsylvania, United States of America, 5 Massachusetts General Hospital, Boston, Massachusetts, United States of America, 6 Mass General Brigham Community Physicians, Somerville, Massachusetts, United States of America

* sldrydenpeterson@bwh.harvard.edu

## Abstract

### Background

The individual risk of severe outcomes following COVID-19 is poorly understood in populations with prior immunity. The lack of contemporary estimates limits support of timely diagnosis and antiviral treatment for individuals most likely to benefit.

### Objective

To determine the risk of severe outcomes following COVID-19 within strata of comorbidities, including patients without documented infection.

### Design

Population-based cohort study utilizing electronic medical records and g methods to account for selection bias in the documentation of COVID-19 illnesses.

### Setting

A large health system in northeastern United States

### Patients

Adults with increased risk conditions (90% vaccinated) and COVID-19 from June to December 2022.

### Measurements

Incidence of composite of inpatient admission within 14 days and death within 28 days of COVID-19 diagnosis.

**Data availability statement:** Statistical code: Available at https://github.com/sldrydenpeter-son/Risk-of-severe-COVID-outcomes. Data set: Data cannot be shared public as includes confidential human subject information. Approval for use of data by researcher should be sought from the Mass General Brigham Institutional Review Board (contact: IRB@partners.org).

**Funding:** This work was made possible with help from the Harvard University Center for AIDS Research, a funded program of the National Institutes of Health (P30 AI060354) and the National Cancer Institute (R01 CA236546). The funders of this work had no role in study design, data collection and analysis, decision to publish, or preparation of the manuscript.

**Competing interests:** The authors have declared that no competing interests exist.

## Results

An estimated 265,248 patients with at least one increased risk condition developed COVID-19, including 76,996 documented cases. Severe outcomes occurred in 3344 (1.3%) patients following COVID-19— 3147 (1.2%) hospitalizations and 376 (0.14%) deaths. In the absence of treatment, individuals with few increased risk conditions (MASS of 3 or less) accounted for 57% of infections and 0.7% developed severe outcomes. In contrast, 2.3% of patients with multiple increased risk conditions (MASS 4 or greater) or severe immunocompromise experienced severe outcomes, including 81% of deaths. The observed risk reduction with antiviral treatment was -0.1% (-0.2 to 0.02%), -0.6% (-0.9 to -0.4%), -1.3% (-2 to -1%), and -1.9% (-3 to -1%) for patients with MASS 3 or less, MASS 4 and 5, MASS 6 or greater, and severe immunocompromise, respectively.

## Limitations

Estimated number COVID-19 cases cannot be directly verified

## Conclusions

Individuals with multiple medical conditions remain at substantial risk for severe outcomes of COVID-19 and benefit from treatment.

## Introduction

Antiviral therapy reduces severe illness from COVID-19 and can avoid destabilizing surges of hospitalizations. Current guidelines from the National Institutes of Health (NIH) [1] support treatment for the majority of the United States adult population [2,3]. Guidance from the World Health Organization. (WHO) strongly supports treatment for patients with COVID-19 at 6% or higher risk hospitalization and offers weak or conditional recommendation for patients at 3% risk of hospitalization [4]. However, evidence to guide treatment decisions is limited regarding the risk of severe COVID-19 outcomes [5] particularly in the context of prevalent immunity from vaccination or prior infection.

Prompt access to ambulatory COVID-19 treatment is resource intensive for outpatient providers and displaces other clinically valuable activities. During periods of increased COVID-19 activity, prescriptions of antiviral treatment out-numbered statins, anti-hypertensives, and other commonly prescribed medications at Mass General Brigham, a large health system in Massachusetts and New Hampshire. Nirmatrelvir with ritonavir– the preferred treatment by the NIH COVID Treatment Panel [1]– has been associated with fatal events related to drug interaction and requires careful modification of chronic medications by providers for many patients to safely utilize it [6]. Alternative infusion-based treatments (e.g., remdesivir) utilize scarce dedicated clinical space and personnel. Additionally, as antivirals transition to the commercial market, costs to patients and their insurers are expected to be substantial [7].

To inform patient outreach efforts, target clinical resources, and educate clinicians, we sought to estimate the risk of subsequent hospitalization and death in strata of comorbidities and treatment receipt following COVID-19 while accounting for gaps in diagnosis and recording of infections.

## Methods

In this retrospective cohort study, we used routinely collected clinical data from a large health system and methodology to account selection bias in case ascertainment to examine the risk

of severe outcomes among increased risk individuals with COVID-19, including the large number of undocumented infections.

## Participants and data

Adult residents (18 years and older) of Massachusetts and New Hampshire with positive laboratory or self-reported and documented direct viral testing from June 1 to December 31, 2022, were included. If a patient had more than one SARS-CoV-2 infection, the most recent infection was used. Only patients with at least one CDC increased risk condition [2] documented were included. The study period was selected to include more than 50,000 recorded cases to facilitate analyses of population subsets in a period with consistent access to outpatient COVID-19 therapeutics. We utilized electronic medical records (EHR) from Mass General Brigham, a nonprofit health system including nine acute care hospitals and a network of ambulatory clinics providing primary and specialty care to an estimated 1.5 million residents of Massachusetts and southern New Hampshire.

Diagnoses, self-reported race and ethnicity, medication and vaccination histories, and dates of hospitalizations and deaths were obtained from the Mass General Brigham COVID-19 Analytics Enclave, an integrated dataset of electronic medical records. Due to integration of Massachusetts and New Hampshire vaccination registries with the Mass General Brigham EHR, doses received in commercial pharmacies and other locations outside of Mass General Brigham are captured. Vaccine receipt documentation is less complete for doses received outside of these states. The patient's neighborhood Area Deprivation Index (ADI) [8] was obtained via geocoding home address to census tract. When census tract ADI was unavailable, zip code or county-averaged ADI was used.

Medical conditions, active medications, and vaccination status present on the day COVID-19 diagnosis were utilized to assess risk strata. For ease of EHR implementation and to align with prior work on effectiveness of treatment [9,10], we utilized the Monoclonal Antibody Screening Score (MASS), a comorbidity index predictive of hospitalization risk [11,12], for comorbidity stratification. Separately, a stratum for severe immunocompromise (organ or stem cell transplantation, hematologic malignancy, or receiving mTOR inhibitors, cyclosporine, mycophenolate, or anti-CD20 therapy) was included to better inform risk discussions and treatment decisions in this vulnerable category of patients. Patients receiving other immunosuppressive therapies for auto-immune disease or other disorders were considered to have moderate immunocompromise.

After completion of the initial analysis, we also described risk by other commonly utilized classification frameworks including the NIH risk group tiers, WHO risk stratification groups, and by age group as supported by CDC guidance [2].

Study investigators had access to the EHR to confirm and update the analytic dataset and scripts directed by data cleaning queries. EMR data as recorded on March 1, 2023 was used for the analysis. Cleaning efforts focused on chronology of diagnosis, treatment, and onset of severe illness. The date of death was updated for two participants, sex was updated for two patients, and height was imputed at the Massachusetts sex-specific mean [13] for 241 participants. Dataset was not updated with information (e.g., testing results, receipt of treatment) recorded only in notes or other unstructured sections of the EHR.

The study was approved by the Mass General Brigham Institutional Review Board and the requirement for informed consent was waived. We followed the STROBE [14] and RECORD [15] guidelines for reporting observational studies of routinely collected health data.

## Epidemic context

During the period studied, the COVID-19 epidemic intensity was moderate with daily reported cases among Mass General Brigham patients ranging from 30 to 60 per 100,000

and 10 to 20 daily admissions to acute care hospitals. Oral antiviral therapy—predominantly nirmatrelvir with ritonavir but also molnupiravir— was available via prescription from ambulatory clinicians. During periods of higher volume, ambulatory surge structures were implemented to preserve access. Outpatient intravenous therapy (initially single-dose bebtelovimab and replaced by multi-dose remdesivir in November 2022) was available in eight locations for patients unable to safely receive nirmatrelvir with ritonavir due to drug-drug interactions or advanced kidney disease. Omicron subvariants BA.2.12.1, BA.5, BQ.1, BQ.1.1, and XBB.1.5 were most prevalent during the study period [16,17].

## Estimation of risk of severe outcomes of COVID-19

A minority of incident COVID-19 cases are recorded [18], creating an analytic challenge to estimate the risk of severe outcome per incident infection. We anticipated improved documentation of infections resulting in outpatient visits or hospitalizations, consequently utilizing all documented infection as used elsewhere [19–22] was expected to overestimate risk of severe outcomes. Alternatively, conditioning on an outpatient diagnosis as utilized in observational studies of SARS-CoV-2 therapeutics [9,23,24] is likely to also introduce bias in risk estimation due to differential access to facility-based testing and to perform and report a positive home test.

Consequently, we utilized approaches to mitigate selection bias [25] to estimate the total number of COVID-19 cases (both recorded and not recorded) that contributed to the observed instances of severe outcomes (hospitalization within 14 days or death within 28 days of positive test). First, we estimated the probability of a preceding outpatient diagnosis (i.e., recorded positive test two or more days prior to admission) for patients with a severe outcome of COVID-19 within strata of *a priori*-selected factors predicting access to timely outpatient diagnosis and severe outcomes. Second, the inverse of these strata-specific probabilities were used as weights to generate a pseudopopulation approximating the total population of patients with COVID-19 at risk for severe outcomes (23), Fig 1.

A binary logistic regression model was used to estimate the probability of a preceding outpatient diagnosis (new positive SARS-CoV-2 molecular or antigen test at least two days prior to any later hospitalization or death) for patients with a severe outcome of COVID-19. Factors included were strata of age (18 to 49, 50 to 64, 65 to 79, and 80 and older), comorbidities (MASS or severe immunocompromise), race and ethnicity (Black, Hispanic/Latinx, and other or White and Asian), high neighborhood disadvantage (census tract in highest quartile of Area Deprivation Index in cohort [8]), immunity status (not fully vaccinated, vaccinated with booster or COVID-19 in past 8 months, or vaccinated with no booster or COVID-19 in past 8 months), and receipt of outpatient COVID-19 antiviral therapy (any of nirmatrelvir-ritonavir, molnupiravir, bebtelovimab, or remdesivir) for the index infection. The resulting strata-specific probabilities were adjusted as suggested by Chapman [26] to reduce small sample bias and yield the nearly unbiased estimator (NUE) [27].

The weighted pseudocohort of estimated increased-risk individuals with incident COVID-19 during the study period was generated with the inverse of the strata-specific probabilities as weights for the participants with an outpatient diagnosis of COVID-19. Participants with a severe outcome of COVID-19 and an outpatient diagnosis were not weighted as diagnosis and outcome directly observed. From the estimated population of individuals with COVID-19 and at least one CDC increased risk condition, we used g-computation to estimate the risk of severe outcomes within strata of comorbidities and immunity status. Robust standard errors were used to estimate confidence limits.

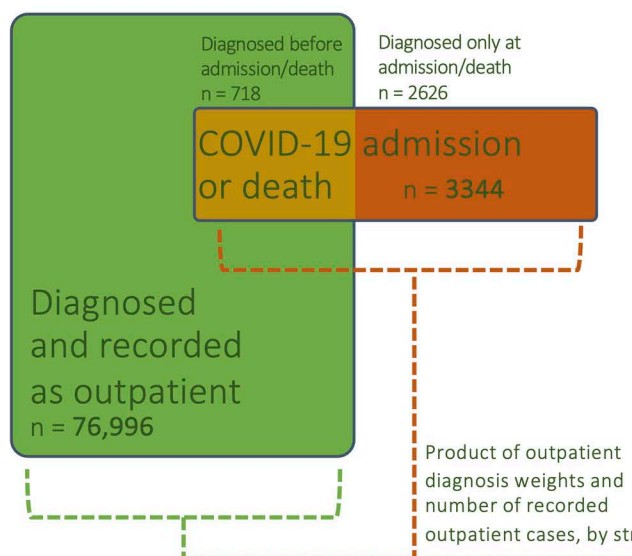

**Step 1.**
Logistic model estimating probabilities of a preceding recorded outpatient diagnosis among adults with severe COVID-19 within strata of age, comorbidities, race/ethnicity, Area Deprivation Index, immunity status, and receipt of outpatient antiviral therapy.

**Step 2.**
Generate weighted pseudocohort of estimated total COVID-19 cases, including undocumented cases, through product documented cases and the strata-specific inverse probability of a recorded outpatient diagnosis

**Step 3.**
Utilizing the generated cohort of all COVID-19 cases among increased risk adults, calculate risk of incident severe outcomes within strata of comorbidities, age, and other factors.

Estimated total COVID-19 cases among increased risk adults
n ~ 265,248

COVID-19 admission or death        n = 3344

**Fig 1. Modeling approach to estimate the total number of incident COVID-19 cases among increased risk patients of Mass General Brigham (June to December 2022) that contributed to the observed hospitalizations and deaths.**

### Estimation of number needed-to-treat to benefit

The number needed-to-treat to prevent an occurrence of a severe outcomes of COVID-19 was calculated as the reciprocal of the risk difference between patients receiving and not-receiving outpatient antiviral treatment. Anticipating bias introduced by differential access and clinician decision to prescribe early outpatient treatment for COVID-19, we utilized the product of the outpatient diagnosis weights and inverse probability weights from a model of outpatient treatment using similar predictors (i.e., age category, comorbidities, race and ethnicity, high ADI, and immunity status). G-estimation was utilized to estimate the expected patient outcomes under conditions of treatment versus non-treatment and corresponding robust standard errors.

### Analytic and model assumptions

Required inference conditions of positivity and consistency are present. However, valid inference from the analytic approach also requires some unverifiable assumptions. First, within strata of covariates the risk of severe outcomes is assumed to be similar. Second, the documentation of SARS-CoV-2 infection is independent of the decision (two or more days later) for inpatient admission. Third, patients with a COVID-19 diagnosis at Mass General Brigham are also admitted at Mass General Brigham if hospitalization is required, or that inflows and outflows from Mass General Brigham balance. Fourth, access to testing, recording of positive tests, and thresholds for admission within strata are consistent throughout the study period. Finally, while analyses conducted are doubly robust [28] the outcome and exposure models share predictors and assume no misspecification of these factors.

All analyses were conducted in R, version 4.3.1 (R Foundation) and analytic code on GitHub (https://github.com/sldrydenpeterson/Risk-of-severe-COVID-outcomes).

## Results

### Recorded COVID-19 cases and severe outcomes

A total of 76,996 Mass General Brigham patients with at least one CDC increased risk criterion had a documented SARS-CoV-2 infection between June 1 and December 31, 2022. During 5671 patient-years of follow-up, 3344 had a severe outcome of COVID-19 due to hospitalization (3147) and/or death (376) within 14 or 28 days of diagnosis, respectively (S1 Fig). While the majority of recorded cases (57%) were among individuals with few comorbidities (MASS 3 or less), patients with MASS comorbidity score of 4 or higher or severe immunocompromise accounted for 68% of all severe outcomes and 81% of all COVID-19 associated deaths.

### Timing of COVID-19 Diagnosis

Patients who developed a severe outcome of COVID-19 were infrequently diagnosed prior to needing hospitalization. An outpatient COVID-19 diagnosis at least two days prior to hospitalization or death was recorded for 718 (21%) patients. A preceding outpatient COVID-19 diagnosis was significantly associated with younger age, being vaccinated, residence in a more disadvantaged neighborhood, and receiving outpatient antiviral treatment. In these adjusted analyses, patient-reported race and ethnicity was not associated with outpatient diagnosis prior to onset of severe COVID-19 illness (S1 Table) [4].

### Estimated total increased risk COVID-19 cases

A total of 265,248 estimated COVID-19 cases (95% confidence interval [CI] 261,620 to 263,624) occurred among increased risk patients during the study period—76,996 recorded and 188,252 unrecorded. The projected actual COVID-19 cases per recorded case was 3.44 and similar across subgroups of patient characteristics (Table 1).

**Table 1. Characteristics of increased risk individuals with COVID-19— recorded and estimated total cases Mass General Brigham, June to December 2022.**

| Characteristic | Recorded cases | Estimated total cases[a] [95% CI] | Ratio of estimated total to recorded cases | Severe outcomes of COVID-19 |
|---|---|---|---|---|
| Total | 76,996 (100%) | 265,248 (100%) [264,215 to 266,281] | 3.44 | 3,344 |
| Age | | | | |
| 18 to 49 | 22,656 (29%) | 74,518 (28%) [73,601 to 75,441] | 3.29 | 638 |
| 50 to 64 | 21,619 (28%) | 71,237 (27%) [70,307 to 72,175] | 3.30 | 558 |
| 65 to 79 | 24,711 (32%) | 90,201 (34%) [89,168 to 91,241] | 3.65 | 1,085 |
| 80 and older | 8,010 (10%) | 29,292 (11%) [28,580 to 30,019] | 3.66 | 1,063 |
| Sex | | | | |
| Female | 47,452 (62%) | 163,873 (62%) [162,829 to 164,913] | 3.45 | 1,827 |
| Male | 29,544 (38%) | 101,375 (38%) [100,335 to 102,419] | 3.43 | 1,517 |
| Race and ethnicity | | | | |
| White | 62,779 (82%) | 215,909 (81%) [215,069 to 216,737] | 3.44 | 2,601 |
| Asian | 3,141 (4.1%) | 10,076 (3.8%) [9,693 to 10,473] | 3.21 | 132 |
| Black | 2,977 (3.9%) | 10,405 (3.9%) [9,993 to 10,834] | 3.50 | 197 |
| Hispanic or Latinx | 4,822 (6.3%) | 16,463 (6.2%) [15,955 to 16,985] | 3.41 | 264 |
| Other or unavailable | 3,277 (4.3%) | 12,396 (4.7%) [11,937 to 12,873] | 3.78 | 150 |
| Increased neighborhood disadvantage (ADI) | 7,541 (9.8%) | 21,813 (8.2%) [21,280 to 22,359] | 2.89 | 378 |
| Vaccination status | | | | |
| Unvaccinated or fewer than 3 doses | 7,788 (10%) | 30,962 (12%) [30,244 to 31,695] | 3.98 | 728 |
| Vaccinated, last dose or COVID-19 ≥ 8 months prior | 43,699 (57%) | 159,390 (60%) [158,355 to 160,423] | 3.65 | 1,859 |
| Vaccinated, last dose or COVID-19 < 8 months prior | 25,509 (33%) | 74,896 (28%) [73,985 to 75,813] | 2.94 | 757 |
| Comorbidity score | | | | |
| MASS 3 or less | 43,960 (57%) | 153,937 (58%) [152,872 to 155,000] | 3.50 | 1,062 |
| MASS 4 and 5 | 10,869 (14%) | 37,173 (14%) [36,424 to 37,935] | 3.42 | 418 |
| MASS 6 or greater | 17,566 (23%) | 62,190 (23%) [61,253 to 63,136] | 3.54 | 1,467 |
| Severe immmunocompromise | 4,601 (6.0%) | 11,948 (4.5%) [11,551 to 12,358] | 2.60 | 397 |
| Body mass index (BMI) | | | | |
| BMI unavailable | 3,454 (4.5%) | 16,578 (6.3%) [16,17,25,149] | 4.80 | 84 |
| BMI less than 25 | 20,234 (26%) | 68,987 (26%) [68,051 to 69,930] | 3.41 | 1,334 |
| BMI 25 to 30 | 26,907 (35%) | 92,710 (35%) [91,695 to 93,731] | 3.45 | 971 |
| BMI 30 to 35 | 15,420 (20%) | 51,890 (20%) [51,052 to 52,737] | 3.37 | 543 |
| BMI greater than 35 | 10,981 (14%) | 35,083 (13%) [34,378 to 35,801] | 3.19 | 412 |
| Immunocompromise | 30,062 (39%) | 100,281 (38%) [99,243 to 101,323] | 3.34 | 1,682 |
| Severe immunocompromise | 4,601 (6.0%) | 11,948 (4.5%) [11,551 to 12,358] | 2.60 | 397 |
| Diabetes | 14,464 (19%) | 47,259 (18%) [46,440 to 48,091] | 3.27 | 1,110 |
| Heart disease or stroke | 12,472 (16%) | 41,718 (16%) [40,926 to 42,522] | 3.34 | 1,300 |
| Pulmonary disease | 5,268 (6.8%) | 17,150 (6.5%) [16,618 to 17,698] | 3.26 | 652 |
| Bipolar, schizophrenia, and other disorders | 2,857 (3.7%) | 9,692 (3.7%) [9,305 to 10,094] | 3.39 | 175 |
| Depression and anxiety | 19,834 (26%) | 65,115 (25%) [64,210 to 66,029] | 3.28 | 809 |
| Hematologic malignancy | 2,915 (3.8%) | 7,623 (2.9%) [7,304 to 7,955] | 2.62 | 263 |
| Solid tumor malignancy | 19,365 (25%) | 66,691 (25%) [65,753 to 67,637] | 3.44 | 1,135 |
| Rheumatologic or inflammatory bowel disease | 9,159 (12%) | 29,386 (11%) [28,731 to 30,055] | 3.21 | 335 |
| Received outpatient COVID antiviral therapy | 28,491 (37%) | 39,551 (15%) [39,050 to 40,058] | 1.39 | 334 |

Abbreviations: ADI, Area Deprivation Index; CI, confidence interval

[a]Cases estimated through accounting for selection bias using the strata-specific, inverse probability of a preceding recorded positive COVID-19 test prior to the onset of a severe outcome as weights.

## Incidence of severe outcomes among all projected COVID-19 cases

Among estimated COVID-19 cases among increased risk individuals, 1.3% of patients (95% CI 1.2 to 1.3%) experienced a severe outcome of either hospitalization (1.2%, 95% CI 1.1 to 1.2%) and/or death (0.14%, 95% CI 0.13% to 0.16%). The risk of severe COVID-19 outcomes varied considerably by strata of comorbidities, vaccination status, and receipt of outpatient antiviral treatment (Table 2). Among vaccinated patients not accessing outpatient COVID-19 treatment, 2.2% of individuals with severe immunocompromise or a MASS comorbidity index of 4 or higher developed severe COVID (95% CI 2.1 to 2.4%) in comparison with 0.59% for increased risk patients with MASS comorbidity index of 3 or lower (95% CI 0.55 to 0.63%).

## Incidence of severe outcomes by NIH, WHO, and age risk stratification guidance

In analyses conducted after completing the primary analysis, risk stratification frameworks by the NIH COVID Treatment Panel [1] (Table 3), the living guidelines of the WHO [4] (Table 4), and by age as supported by CDC [2] (Table 5) all classified patients not accessing antiviral treatment into sequential categories of risk of severe outcomes (Fig 2). In contrast with the primary analysis, the NIH and WHO frameworks include both patients with moderate and severe immunocompromise in the highest risk group. In a model controlling bias by age,

**Table 2. Risk of severe COVID-19 by comorbidity, immunity, and outpatient treatment status— Mass General Brigham, June to December 2022.**

**Hospitalization and/or death risk (95% CI)[a]**

| MASS score[b] | Vaccination status | No outpatient COVID-19 antiviral[c] | Prescribed outpatient COVID-19 antiviral | Adjusted risk difference and number needed to treat to benefit (95% CI)[d] |
|---|---|---|---|---|
| Severe immunocompromise | Unvaccinated or fewer than 3 doses | 6.4% (4% to 9%) | 2.1% (0.5% to 4%) | RD -1.9% (-3% to -1%) NNTB 52 (39 to 78) |
| | Vaccinated, last dose or COVID-19 ≥ 8 months prior | 2.5% (2% to 3%) | 1.2% (0.6% to 2%) | |
| | Vaccinated, last dose or COVID-19 < 8 months prior | 2.2% (1% to 3%) | 1.2% (0.5% to 2%) | |
| MASS 6 or greater | Unvaccinated or fewer than 3 doses | 5.6% (3% to 8%) | 1.4% (0.4% to 2%) | RD -1.3% (-2% to -1%) NNTB 80 (66 to 100) |
| | Vaccinated, last dose or COVID-19 ≥ 8 months prior | 2.2% (1% to 3%) | 0.85% (0.6% to 1%) | |
| | Vaccinated, last dose or COVID-19 < 8 months prior | 1.9% (0.9% to 3%) | 0.83% (0.5% to 1%) | |
| MASS 4 and 5 | Unvaccinated or fewer than 3 doses | 2.1% (1% to 3%) | 0.54% (0.08% to 1%) | RD -0.63% (-0.9% to -0.4%) NNTB 160 (120 to 250) |
| | Vaccinated, last dose or COVID-19 ≥ 8 months prior | 0.82% (0.6% to 1%) | 0.32% (0.1% to 0.5%) | |
| | Vaccinated, last dose or COVID-19 < 8 months prior | 0.72% (0.5% to 1%) | 0.31% (0.1% to 0.5%) | |
| MASS 3 or less | Unvaccinated or fewer than 3 doses | 1.5% (1% to 2%) | 0.5% (0.09% to 0.9%) | RD -0.11% (-0.2% to 0.02%) NNTB 900 (420 to Inf) |
| | Vaccinated, last dose or COVID-19 ≥ 8 months prior | 0.59% (0.5% to 0.7%) | 0.3% (0.1% to 0.5%) | |
| | Vaccinated, last dose or COVID-19 < 8 months prior | 0.52% (0.4% to 0.6%) | 0.29% (0.1% to 0.5%) | |

Abbreviations: CI, confidence interval; NNTB, number needed to treat to benefit; RD, risk difference

[a]Hospitalization within 14 days or death within 28 days of COVID-19 diagnosis.

[b]Monoclonal antibody screening score (MASS) calculated as age 65 years and older (2 points), BMI 35 and higher (2), diabetes (2), chronic kidney disease (3), cardiovascular disease in a patient 55 years and older (2), chronic respiratory disease in a patient 55 years and older (3), hypertension in a patient 55 years and older (1), and immunocompromised status (3). Risk for individuals with severe immunocompromise— organ or stem cell transplantation, hematologic malignancy, or receiving mTOR inhibitors, cyclosporine, mycophenolate, or anti-CD20 therapy— estimated separately.

[c]Outpatient treatments during the study period included oral nirmatrelvir-ritonavir (90%), oral molnupiravir (2%), intravenous bebtelovimab (5%), and intravenous remdesivir (3%).

[d]Strata-specific estimates calculated through marginal inverse-probability weighted models to reduce expected bias in access and acceptance to outpatient treatment by vaccination status, race and ethnicity, neighborhood disadvantage, and age.

**Table 3. Risk of severe COVID-19 by NIH prioritization tier for increased risk patients not accessing antiviral treatment— Mass General Brigham, June to December 2022.**

| Risk group | Projected cases (95% CI) | Hospitalizations[a] | Deaths[a] | Covid Hosp or Death (95% CI) |
|---|---|---|---|---|
| **Tier 1:** Individuals with moderate or severe immunocompromise, unvaccinated and 75 or older, or unvaccinated and 65 and or older with additional risk factors | 46,322 (21%) [45,455 to 47,188] | 1,077 | 138 | 2.4% [2.2 to 2.5%] |
| **Tier 2:** Unvaccinated individuals not included in Tier 1 who are 65 or older or have clinical risk factors | 11,273 (5%) [10,824 to 11,721] | 211 | 5 | 1.7% [1.4 to 2.0%] |
| **Tier 3:** Vaccinated individuals not included in Tier 1 who are 65 or older or have clinical risk factors | 168,102 (74%) [167,107 to 169,097] | 1,569 | 181 | 0.95% [0.90 to 1.0%] |

Abbreviation: CI, confidence interval

[a]Hospitalization within 14 days or death within 28 days of COVID-19 diagnosis.

**Table 4. Risk of severe COVID-19 by WHO risk groups for increased risk patients not accessing antiviral treatment— Mass General Brigham, June to December 2022.**

| WHO risk group | Projected cases (95% CI) | Hospitalizations[a] | Deaths[a] | Covid Hosp or Death (95% CI) |
|---|---|---|---|---|
| **High risk:** People who have been diagnosed with immunodeficiency syndromes, undergone a solid organ transplant and receiving immunosuppressants, or have autoimmune disease and are receiving immunosuppressants | 39,981 (18%) [39,181 to 40,781] | 848 | 97 | 2.2% (2.0% to 2.4%) |
| **Moderate risk:** People over 65 years old or with certain health conditions: obesity, diabetes, chronic cardiopulmonary disease, chronic kidney disease, chronic liver disease, active cancer, disability, and comorbidities of chronic disease | 148,947 (66%) [147,844 to 150,051] | 1,808 | 221 | 1.1% (1.0% to 1.2%) |
| **Low risk:** People who are not high or moderate risk | 36,768 (16%) [36,053 to 37,484] | 201 | 6 | 0.40% (0.31% to 0.49%)[b] |

Abbreviation: CI, confidence interval

[a]Hospitalization within 14 days or death within 28 days of COVID-19 diagnosis.

[b]Estimates only includes individuals with at least one increased risk condition. Consequently, estimated risk likely higher than if individuals without increased risk conditions were included.

**Table 5. Risk of severe COVID-19 by age group for increased risk patients not accessing antiviral treatment— Mass General Brigham, June to December 2022.**

| Age group | Projected cases (95% CI) | Hospitalizations[a] | Deaths[a] | Covid Hosp or Death [95% CI] |
|---|---|---|---|---|
| 85 and older | 11,651 (5%) [11,143 to 12,159] | 553 | 120 | 4.9% [4.3% to 5.4%] |
| 75 to 84 | 32,131 (14%) [31,331 to 32,930] | 674 | 85 | 1.9% [1.7% to 2.1%] |
| 65 to 74 | 54,966 (24%) [53,993 to 55,940] | 586 | 68 | 1.1% [0.94% to 1.2%] |
| 50 to 64 | 59,846 (27%) [58,936 to 60,756] | 467 | 38 | 0.67% [0.58% to 0.76%] |
| 30 to 49 | 50,321 (22%) [49,545 to 51,096] | 437 | 9 | 0.69% [0.59% to 0.79%] |
| 18 to 29 | 16,782 (7%) [16,285 to 17,280] | 140 | 4 | 0.55% [0.39% to 0.70%] |

Abbreviation: CI, confidence interval

[a]Hospitalization within 14 days or death within 28 days of COVID-19 diagnosis.

## Hospitalization or death following COVID-19 infection

Increased-risk individuals without outpatient antiviral therapy— 236,757 total infections (recorded and unrecorded)

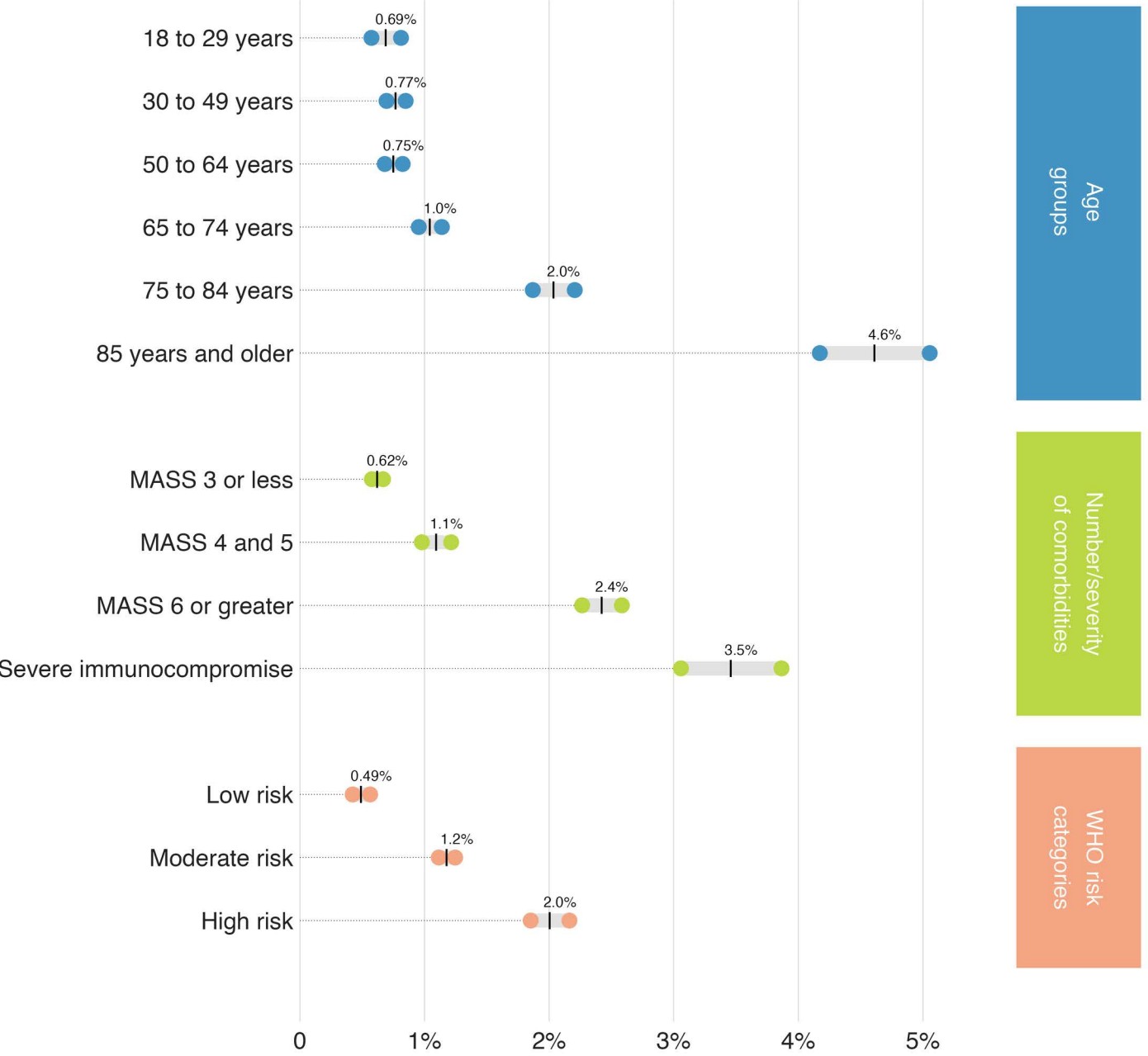

**Fig 2. Estimated risk of hospitalization and/or death among nonhospitalized adults with COVID-19 by age and comorbid conditions.** Confidence limits (95%) shown. Population includes all estimated infections (June to December 2022) occurring among people 18 years and older with one or more risk factor associated with increased risk of complications [2]. Monoclonal antibody screening score (MASS) [11,12] calculated as age 65 years and older (2 points), BMI 35 and higher (2 points), diabetes (2 points), chronic kidney disease (3 points), cardiovascular disease in a patient 55 years and older (2 points), chronic respiratory disease in a patient 55 years and older (3 points), hypertension in a patient 55 years and older (1 points), and immunocompromised status (3 points). Risk for individuals with severe immunocompromise— organ or stem cell transplantation, hematologic malignancy, or receiving mTOR inhibitors, cyclosporine, mycophenolate, or anti-CD20 therapy— estimated separately. World Health Organization (WHO) risk categories include: high risk(people who have been diagnosed with immunodeficiency syndromes, undergone a solid organ transplant and receiving immunosuppressants, or have autoimmune disease and are receiving immunosuppressants), moderate risk (people over 65 years or with certain health conditions: obesity, diabetes, chronic cardiopulmonary disease, chronic kidney disease, chronic liver disease, active cancer, disability, and comorbidities of chronic disease), and low risk (people who are not high or moderate risk) [4].

neighborhood ADI, race and ethnicity, and vaccination status, the estimated risk for severe outcomes of COVID-19 was 0.80% (95% CI 0.65 to 0.96) and 2.5% (1.6 to 3.3%) for patients with moderate and severe immunocompromise, respectively.

### Antiviral treatment and risk reduction

Outpatient COVID-19 treatment was prescribed to 28,491 patients— nirmatrelvir-ritonavir (90%, 25,772), bebtelovimab (5%, 1486), outpatient remdesivir (3%, 881), molnupiravir (2%, 565). Some patients were prescribed more than one treatment regimen. Treatment was more likely to be prescribed to patients with more comorbidities, multidose vaccination, residence in a less disadvantaged neighborhood, and self-reported non-Hispanic White or Asian race and ethnicity.

In adjusted analyses, antiviral treatment was associated with reduction in risk of severe outcomes of COVID-19 in all comorbidity strata. However, the among patients with MASS 3 or less the trend towards lower risk was not significant. The estimated number needed to treat to prevent a severe outcome of COVID-19 range from 52 for patients with severe immunocompromise to 900 for patients with MASS score or 3 or less.

## Discussion

In this analysis of COVID-19 outcomes among increased risk individuals from a large health system, patients with incomplete vaccination, multimorbidity, and severe immunocompromise experienced the highest risk (estimated 2 to 6%) of hospitalization and death. Vaccinated patients meeting increased risk criteria but with few comorbidities, experienced more modest risk (estimated 0.5%) of severe outcomes. These findings directed prescribing and outreach to Mass General Brigham patients at the highest risk of severe outcomes (i.e., greater than 2%) to support completion of vaccination, prompt symptomatic and post-exposure testing, and an individualized plan to access an appropriate treatment if they developed COVID-19.

To our knowledge, the current analysis is the first in the Omicron era to estimate the probability of severe outcomes following COVID-19 within risk strata. Estimates of risk of severe outcomes in the absence of treatment are available from control arms of therapeutic clinical trials conducted among nonhospitalized adults at increased risk. Trial participants were unvaccinated, and studies were conducted prior to emergence of Omicron variants. Severe outcomes occurred in 4.6% to 9.7% of placebo recipients in registration trials of monoclonal antibodies [29–31] and antiviral agents [32–34]. Current WHO guidance utilizes similar estimates from prior, higher risk periods of the COVID-19 pandemic [4,5]. The findings of our analysis Mass General Brigham data suggest that these estimates may inflate contemporary risk of severe outcomes.

This work builds on strong observational data. Conducted contemporaneously in the United Kingdom, investigators utilized health records for vaccinated individuals and similarly identified that increasing number of comorbidities, age over 65, and longer duration from vaccination were associated with increased risk of severe COVID-19 outcomes [35]. In a separate analysis utilizing similar resources, under vaccination remained strongly associated with increased risk of hospitalization or death despite high prevalence of prior SARS-CoV2 infection [36]. However, these analyses could only estimate relative risks among those with recorded infection and not estimate the probability of severe outcomes as the underlying denominator of community COVID-19 cases leading to hospitalizations was not known.

Similar to prior estimates [18,37], the majority (estimated 71%) of individuals with COVID-19 and factors increasing risk for severe outcomes were undiagnosed or had unreported infection. Decreased detection among the increased risk population— particularly among older individuals and unvaccinated individuals of all ages— remains a primary barrier

to effective treatment of COVID-19. Additionally, disparities were evident in the receipt of treatment among those with a reported diagnosis with decreased access for the youngest and oldest age groups, unvaccinated individuals, and residents of disadvantaged neighborhoods. Further clinical innovation and research is needed to improve access to timely diagnosis and effective treatment for COVID-19.

Vaccinated individuals with few comorbid conditions (MASS 3 or less) received nearly half (49%) of all antiviral treatment courses for COVID-19. The observed risk of severe outcomes without treatment was modest in these patients and the estimated risk reduction through treatment low. These findings should be confirmed in other contexts but suggest that revision to treatment recommendations be considered. A focus on treatment of patients with multi-morbidity (MASS 4 or higher) or incomplete vaccination is expected to improve alignment of scarce clinical resources and potential for harm, particularly among individuals with potential drug-drug interactions with nirmatrelvir-ritonavir. Additionally, programs to improve testing (including reporting of positive home tests) and timely access to treatment are needed for this high-risk group.

The findings of these analyses should be considered along with several limitations. Inference from the weighted analytic models accounting for unrecorded infections relies on unverifiable assumptions that may be partially violated (i.e., knowledge of COVID-19 diagnosis may alter hospitalization decisions several days later). The analyses rely on routinely-collected data and uneven data completeness could introduce misclassification bias. For example, COVID-19 treatments outside of Mass General Brigham (e.g., treatment received in nursing homes or other congregate care settings) are not captured in our analyses and could bias risk estimates downwards. Due to challenges of cause attribution, all hospitalizations within two weeks and deaths within four weeks were included as severe outcomes; however, COVID-19 may not have played an important causal role in each of these endpoints. Importantly, comparisons of risk for individuals utilizing and not utilizing outpatient COVID-19 therapy are expected to have residual bias by factors affecting treatment access and decisions, such as the severity of comorbidities and COVID-19 symptoms, concurrent medications, and social determinants of health. Vaccination protection may be underestimated, as an Omicron-specific vaccine only became available late during the study period and had low uptake. The stratification framework implemented at Mass General Brigham included four categories of comorbidity burden and three categories of vaccination status. Simpler risk stratification frameworks, such as those by the WHO or by age alone, may be easier to communicate and implement. Finally, these data are drawn from an affluent metropolitan region in the United States and may not be generalizable to other contexts.

This study provides estimates of risk of severe outcomes of COVID-19 among individuals with high levels of immunity from vaccination and prior infection. Severity of illness is lower among individuals with increased risk conditions than during prior periods of the pandemic [38,39], however greater than 2% of patients who have severe immunocompromise, or considerable multimorbidity (MASS 6 or greater) require hospitalization following COVID-19. Supporting vaccination completion and ensuring prompt access to treatment among these highest risk individuals should be prioritized to reduce ongoing hospitalizations and deaths from COVID-19.

## Supporting information

**S1 Fig. Cohort diagram.**
(PDF)

**S1 Table. Odds ratios of preceding outpatient diagnosis of COVID-19 among patients with severe outcomes by selected characteristics.**
(DOCX)

## Author contributions

**Conceptualization:** Scott Dryden-Peterson, Arthur Y. Kim, John Fangman, Lindsey R. Baden, Ann E. Woolley.

**Data curation:** Scott Dryden-Peterson, Andy Kim, Mary-Ruth Joyce, David Rubins, Lindsey R. Baden, Ann E. Woolley.

**Formal analysis:** Scott Dryden-Peterson, Ellen C. Caniglia, Ann E. Woolley.

**Funding acquisition:** Scott Dryden-Peterson.

**Investigation:** Scott Dryden-Peterson.

**Methodology:** Scott Dryden-Peterson, Ellen C. Caniglia, David Rubins, Arthur Y. Kim, John Fangman, Ann E. Woolley.

**Resources:** Ann E. Woolley.

**Supervision:** Scott Dryden-Peterson, Ann E. Woolley.

**Writing – original draft:** Scott Dryden-Peterson, Ann E. Woolley.

**Writing – review & editing:** Scott Dryden-Peterson, Andy Kim, Ellen C. Caniglia, Mary-Ruth Joyce, David Rubins, Arthur Y. Kim, John Fangman, Lindsey R. Baden, Ann E. Woolley.

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
