## [Decision Letter · Decision Letter 0]

12 Nov 2024

PONE-D-24-42702Severe outcomes of COVID-19 among adults with increased risk conditions: a population-based observational studyPLOS ONE

Dear Dr. Dryden-Peterson,

Thank you for submitting your manuscript to PLOS ONE. After careful consideration, we feel that it has merit but does not fully meet PLOS ONE’s publication criteria as it currently stands. Therefore, we invite you to submit a revised version of the manuscript that addresses the points raised during the review process.

Editor comments: It is noted that there are only four references are cited in Discussion. The discussion lacks comparison and correlation with others' results. Further discussion of your results will better improve the equality of your work.

Please submit your revised manuscript by Dec 27 2024 11:59PM. If you will need more time than this to complete your revisions, please reply to this message or contact the journal office at plosone@plos.org . Please include the following items when submitting your revised manuscript:

We look forward to receiving your revised manuscript.

Kind regards,

Benjamin M. Liu, MBBS, PhD, D(ABMM), MB(ASCP)

Academic Editor

PLOS ONE

Journal Requirements:

2. Thank you for stating the following financial disclosure: [This work was made possible with help from the Harvard University Center for AIDS Research, a funded program of the National Institutes of Health (P30 AI060354) and the National Cancer Institute (R01 CA236546).]. Please state what role the funders took in the study. If the funders had no role, please state: "The funders had no role in study design, data collection and analysis, decision to publish, or preparation of the manuscript." If this statement is not correct you must amend it as needed. Please include this amended Role of Funder statement in your cover letter; we will change the online submission form on your behalf.

4. We note you have included a table to which you do not refer in the text of your manuscript. Please ensure that you refer to Table 2-5 in your text; if accepted, production will need this reference to link the reader to the Table.

5. Please include captions for your Supporting Information files at the end of your manuscript, and update any in-text citations to match accordingly. Please see our Supporting Information guidelines for more information: http://journals.plos.org/plosone/s/supporting-information .

Reviewers' comments:

Reviewer's Responses to Questions

**Comments to the Author**

1. Is the manuscript technically sound, and do the data support the conclusions?

Reviewer #1: Yes

2. Has the statistical analysis been performed appropriately and rigorously? 

Reviewer #1: N/A

3. Have the authors made all data underlying the findings in their manuscript fully available?

Reviewer #1: Yes

4. Is the manuscript presented in an intelligible fashion and written in standard English?

Reviewer #1: Yes

5. Review Comments to the Author

Reviewer #1: Methodology:

It’s helpful that the authors provided context on the COVID-19 situation during the study period: "During the period studied, COVID-19 epidemic intensity was moderate, with daily reported cases among Mass General Brigham patients ranging from 30 to 60 per 100,000 and 10 to 20 daily admissions to acute care hospitals. The prevalent Omicron subvariants during this time included BA.2.12.1, BA.5, BQ.1, BQ.1.1, and XBB.1.5." This situational background aids reader understanding.

To reduce selection bias and estimate the total number of cases, the authors used a binary logistic regression model or “g-computation.” Including a workflow diagram would further clarify this process for readers.

The estimation model itself is interesting, and I appreciate that the authors made it available on GitHub. However, it seems to lack a "description" file, which prevented me from installing it. Adding an R Markdown file would be beneficial, as it would provide an easier-to-follow explanation of the process, saving readers the effort of working it out themselves.

Results:

Since the authors are using R, enhancing the data illustration with more effective visualizations could provide clearer insights than simple charts.

In Table 1, including a statistical test (t-test or ANOVA) to indicate significant differences in severe outcomes across groups would improve the data’s clarity and interpretation.

The authors note that Omicron variants were prevalent during the study period, and it is known that vaccine effectiveness against Omicron is reduced. This compromises the robustness of subgroup analyses based on vaccination status. Additionally, the effectiveness of vaccination in severely immunocompromised patients remains uncertain, further complicating conclusions drawn from this study. I recommend clearly addressing these limitations in the discussion section.

Writing:

“Background: The risk of severe outcomes following COVID-19 is poorly understood in

populations with prior immunity limiting efforts to facilitate diagnosis and treatment for

individuals most likely to benefit.”

--This sentence leaves me a headache!!!.

6. PLOS authors have the option to publish the peer review history of their article (what does this mean? ). If published, this will include your full peer review and any attached files.

**Do you want your identity to be public for this peer review?** For information about this choice, including consent withdrawal, please see our Privacy Policy .

Reviewer #1: No

---

## [Author Response · Author response to Decision Letter 0]

18 Nov 2024

Dear Editors,

We continue to appreciate the careful review that our manuscript examining the effectiveness of nirmatrelvir plus ritonavir in preventing severe COVID-19 disease.

Please find below detailed responses to the suggestions and concerns raised by editor and reviewer:

Academic Editor

1. It is noted that there are only four references are cited in Discussion. The discussion lacks comparison and correlation with others' results. Further discussion of your results will better improve the equality of your work.

Thank you for this suggestion. We have substantially expanded the discussion to contextualize findings within existing studies (eg, randomized trials of therapeutics) and a modeling that forms the basis of estimates suggested in the WHO guidance. Additionally, we review contemporary work on increased risk and highlight the need for estimates of probability of severe outcomes.

Reviewer #1

1. To reduce selection bias and estimate the total number of cases, the authors used a binary logistic regression model or “g-computation.” Including a workflow diagram would further clarify this process for readers.

We agree that this would be a good addition and have added Figure 1. We appreciate this suggestion.

2. The estimation model itself is interesting, and I appreciate that the authors made it available on GitHub. However, it seems to lack a "description" file, which prevented me from installing it. Adding an R Markdown file would be beneficial, as it would provide an easier-to-follow explanation of the process, saving readers the effort of working it out themselves.

We will update the GitHub repository to provide a guide.

3. Since the authors are using R, enhancing the data illustration with more effective visualizations could provide clearer insights than simple charts.

We have created Figure 2 that summarizes the key findings and could be used as a ‘striking image’.

4. In Table 1, including a statistical test (t-test or ANOVA) to indicate significant differences in severe outcomes across groups would improve the data’s clarity and interpretation.

Table 1 is intended to describe the cohort and outcomes. The comparisons within characteristics do not have a causal interpretation or specific testable hypotheses. We feel that adding a statistic test would be misleading.

5. The authors note that Omicron variants were prevalent during the study period, and it is known that vaccine effectiveness against Omicron is reduced. This compromises the robustness of subgroup analyses based on vaccination status. Additionally, the effectiveness of vaccination in severely immunocompromised patients remains uncertain, further complicating conclusions drawn from this study. I recommend clearly addressing these limitations in the discussion section.

The goal of the analysis is not to assess effectiveness of vaccination, but rather to describe probability of severe outcomes within clinical parameters. However, we agree that it is important to highlight that the Omicron-specific vaccine was underutilized due to timing of its release. This has been added to discussion as limitation.

6. “Background: The risk of severe outcomes following COVID-19 is poorly understood in

populations with prior immunity limiting efforts to facilitate diagnosis and treatment for

individuals most likely to benefit.”--This sentence leaves me a headache!!!.

We have updated this section to improve readability.

Journal Requirements:

We believe that the manuscript adheres to this guidance.

2. Thank you for stating the following financial disclosure: [This work was made possible with help from the Harvard University Center for AIDS Research, a funded program of the National Institutes of Health (P30 AI060354) and the National Cancer Institute (R01 CA236546).]. Please state what role the funders took in the study. If the funders had no role, please state: "The funders had no role in study design, data collection and analysis, decision to publish, or preparation of the manuscript." If this statement is not correct you must amend it as needed. Please include this amended Role of Funder statement in your cover letter; we will change the online submission form on your behalf.

This statement is correct and the cover letter has been updated.

This statement is present in the Methods section.

4. We note you have included a table to which you do not refer in the text of your manuscript. Please ensure that you refer to Table 2-5 in your text; if accepted, production will need this reference to link the reader to the Table.

Thank you for noting that we had abbreviated Table. We have corrected this in the updated manuscript and added references to two new Figures suggested during peer review.

No supporting information other than tables and figures included in the text.

Reference list confirmed to be complete. No cited work is known to have been retracted.

Again, we appreciate the through and prompt review. Thank you.

---

## [Editor Report · Decision Letter 1]

13 Dec 2024

Severe outcomes of COVID-19 among adults with increased risk conditions: a population-based observational study

PONE-D-24-42702R1

Dear Dr. Dryden-Peterson,

We’re pleased to inform you that your manuscript has been judged scientifically suitable for publication and will be formally accepted for publication once it meets all outstanding technical requirements.

Kind regards,

Benjamin M. Liu, MBBS, PhD, D(ABMM), MB(ASCP)

Academic Editor

PLOS ONE
---

## [Editor Report · Acceptance letter]

PONE-D-24-42702R1

PLOS ONE

Dear Dr. Dryden-Peterson,

I'm pleased to inform you that your manuscript has been deemed suitable for publication in PLOS ONE. Congratulations! Your manuscript is now being handed over to our production team.

Kind regards,

on behalf of

Dr. Benjamin M. Liu

Academic Editor

PLOS ONE